# A Study on Aqueous Dispersing of Carbon Black Nanoparticles Surface-Coated with Styrene Maleic Acid (SMA) Copolymer

**DOI:** 10.3390/polym14245455

**Published:** 2022-12-13

**Authors:** Jaeseon Lee, Jihyun Bae, Woonjung Kim, Seungho Lee

**Affiliations:** Department of Chemistry, Hannam University, Daejeon 305-811, Republic of Korea

**Keywords:** carbon black (CB) dispersing, dispersion stability, styrene maleic acid copolymer, particle size distribution, asymmetrical flow field-flow fractionation (AsFlFFF)

## Abstract

Carbon black (CB) particles tend to aggregate in aqueous solutions, and finding an optimum dispersing condition (e.g., selection of the type of dispersant) is one of the important tasks in related industries. In the present study, three types of styrene maleic acid (SMA) copolymer dispersants were synthesized, labeled respectively ‘SMA-1000’, ‘SMA-2000’, and ‘SMA-3000’, which have 1, 2, and 3 styrene groups in their repeating units. Then, asymmetrical flow field-flow fractionation (AsFlFFF) was employed to measure the particle size distributions of the aqueous CB dispersions. For the particle size analysis of the CB dispersions, dynamic light scattering (DLS) showed relatively lower reproducibility than AsFlFFF. AsFlFFF showed that the use of SMA-3000 yielded a CB dispersion with the most uniform particle size distribution. When the SMA-3000 dispersant was used, the particle size tended to increase after 1 h of milling as the milling time increased, probably due to the re-agglomeration of the particles by excessive milling. The particle size distributions from AsFlFFF were consistent with the colorimetric observations. With the SMA-3000 dispersant, the lowest $L^{*}$ value was observed after 1 h of milling. The AsFlFFF and colorimetric analyses suggest that a stable CB dispersion can be obtained by either 3-h of milling with the SMA-2000 or 1-h of milling with the SMA-3000.

## 1. Introduction

Carbon black (CB) shows excellent colorability, electrical conductivity, weather resistance, and chemical resistance. It has a broad range of industrial applications as a material-reinforcing agent [1] or as a filler for plastics or elastomers [2]. CB is also widely used as a black pigment or as a shading agent for black matrixes [3,4,5]. 

High-viscosity (or high concentration) CB dispersions are widely used in a variety of industries, including cosmetics, rubbers, tires, plastics, and color filter inks for flat panel display (FPD). CB has some restrictions on industrial applications due to its tendency to form agglomerates owing to strong inter-particle affinity, which results in poor dispersion stability. Being hydrophobic, CB has poor wettability, making its dispersion at high concentrations in aqueous systems difficult [6,7,8,9].

Figure 1 shows three stages of a typical CB-dispersing procedure [10]. The stage-1 is a wetting stage, where the air and moisture present on the surface of the particles are replaced by the solution. The solution penetrates into empty spaces in the agglomerates. In stage 2, the CB agglomerates are mechanically ground, during which some of the particles may re-agglomerate. Stage 3 is the stabilization stage, where the CB dispersion is treated to prevent precipitation or re-agglomeration of the particles [11,12].

Effective dispersing of the particles and, thus, stabilization of the dispersion relies on electrostatic repulsion between the charged particles in the solution [13,14]. Moreover, a layer adsorbed on the surface of particles can provide steric hindrance [15,16,17,18] to maintain a distance between particles in order to prevent them from agglomerating [19,20].

Figure 2 illustrates three components of an aqueous CB dispersion (CB particles, solvent (water), and dispersant) and the types of interactions among them that affect dispersing of CB particles. The degree of hydration of the CB particle surface affects the interfacial tension. The solubility is affected by the type and composition of the dispersant in water. To ensure stable dispersion, it is important to maintain a balance between the three components [10,21].

Since the surface of carbon black particles is hydrophobic, CB particles are not well dispersed in water. As shown in Figure 2, a dispersant is usually required to disperse the CB particles in water. Dispersants, such as surfactants, have both hydrophobic and hydrophilic groups. The dispersibility of a dispersant depends on the type of the dispersant, which affects the affinity between the dispersant and the CB particles. As the affinity increases, dispersants are adsorbed more strongly on the surface of the CB particles and will yield a more stable CB dispersion. Since the surface of CB is hydrophobic, the hydrophobic portion of the dispersant is strongly coated to show more stable dispersibility.

Generally, dispersants are divided into two types, low molecular weight (LMW) and high molecular weight types (HMW) [22]. The LMW dispersants usually lack effectiveness on organic particles consisting of carbon, hydrogen, oxygen, and nitrogen atoms. Moreover, LMW dispersants may not provide sufficient steric hindrance even if they can adsorb to the surface of CB particles. Unlike LMW dispersants, HMW dispersants can provide sufficient steric hindrance for particle dispersion. HMW dispersants are generally used to disperse CB particles, which usually have a group with side chains that can be adsorbed on the surface of CB particles. For dispersion stability, the dispersant molecules must be strongly attached to the surface of the CB particles, and sufficient steric hindrance must exist between the CB particles.

The optimization of the particle-dispersing process includes the selection of proper dispersants, the amount of dispersant, milling speed, milling time, and the development of dispersing method. Among them, selecting an appropriate dispersant for an aqueous dispersion of CB particles or finding optimal dispersion conditions (e.g., mechanical dispersion time) is not straightforward. If the mechanical dispersion time (e.g., milling time) is not sufficient, the particle size distribution tends to be broad due to the presence of agglomerated particles. Conversely, if the milling time is too long (‘over-dispersing’), the specific surface area of the CB particles increases as the particle size decreases, which increases the van der Waals attraction between the CB particles, leading to agglomeration of the particles.

The dispersion’s viscosity and particle size distribution are two parameters that are usually considered for optimizing the particle dispersing process. Therefore, there is a need to develop an accurate and reproducible analytical method for determining the viscosity and particle size distribution of CB dispersions.

Viscosity is often measured to test the stability of CB dispersions. However, sometimes the difference in viscosity of CB dispersions prepared by different methods is not large enough to be accurately measured unless measured over several weeks at elevated temperatures.

Another (and maybe more direct) method for testing the stability of CB dispersion is to measure the particle size distribution of the dispersion. Dynamic light scattering (DLS) has been widely used for particle size measurements [23,24]. DLS is relatively simple to use and generally shows good reproducibility for particle size distribution analysis for low-viscosity dispersions. However, for high-viscosity dispersions, the reproducibility of DLS results tends to decrease due to low particle mobility at relatively high concentrations [25].

In this study, styrene-maleic acid (SMA)-based dispersants, a type of HMW dispersants known to have excellent thermal stability, were synthesized. Then, CB dispersions were prepared through mechanical milling [26]. The prepared SMA dispersant was strongly coated on the surface of the CB particles, and it was confirmed whether sufficient steric hindrance existed between the CB particles. For confirmation, various techniques, including DLS and asymmetric flow field-flow fractionation (AsFlFFF) [27], have been used to analyze the prepared CB dispersions and to find the optimal CB dispersion conditions (type of dispersant and dispersion time, etc.). In addition, thermal stability was evaluated to confirm the stability of the prepared CB dispersion.

## 2. AsFlFFF Theory

As a member of the field-flow fractionation (FFF) family, AsFlFFF provides a separation of dispersed particles passing through a hollow channel according to their diffusion coefficients (and thus their particle sizes). As shown in Figure 3, AsFlFFF separates dispersed particles according to particle size by eluting the particles in order of particle size. [28].

One of the unique features of AsFlFFF is that it allows the direct determination of the hydrodynamic diameter ($d_{H}$) of the particles from the measured retention time ($t_{R}$) of the particles by [29,30,31,32].
(1)$$d_{H}=\frac{2kTV^{0}}{\pi \eta w^{2}V_{c}t^{0}}t_{R}$$

In Equation (1), k is the Boltzmann constant; T is the absolute temperature (K); $V^{0}$ is the void volume of the AsFlFFF channel; η is the viscosity of the carrier liquid; w is the channel thickness; $V_{c}$ is the cross-flow rate, and $t^{0}$ is the time required to pass through the channel volume (‘void time’). All parameters in Equation (1), except $t_{R}$, are constant under a given experimental condition. Thus, using Equation (1), the particle size and thus the particle size distribution of a particle dispersion can be determined by measuring $t_{R}$ in AsFlFFF [10,33,34]

## 3. Experimental

### 3.1. Materials

CB particles with a surface area of 75 m^2^/g and a bulk density of 80–120 g/L (Alfa Aesar, Haverhill, MA, USA) was used to prepare the CB dispersions. For the synthesis of the dispersants, deionized (DI) water (Milli Q PLUS) and styrene maleic anhydride (SMAnh) powder (Sartometer Co., Ltd., Exton, PA, USA) were used, along with ammonia water (25–28%, Duksan, Republic of Korea). FL-70 (Fisher Chemical, Fair Lawn, NJ, USA) and sodium azide (Sigma-Aldrich, St. Louis, MO, USA) were used to prepare the AsFlFFF carrier liquid.

### 3.2. Instruments

An overhead stirrer (HT-50DX, Daihan, Seoul, Republic of Korea) was used to prepare the CB dispersions in a heating mantle (DH-WHM12054-EA, Daihan, Seoul, Republic of Korea). The conductivity and viscosity of the CB dispersions were measured using a conductivity meter (OHAUS 3100C, OHAUS Corp, Seoul, Republic of Korea) and a viscometer (Brookfield, LVDVE 230, Ct Vernon Hills, IL, USA), respectively. A colorimeter (CR-400 Choma Meter, Konica Minolta, Tokyo, Japan) was used to measure the color of the CB dispersions.

For the determination of particle size and particle size distribution of CB dispersions, a DLS (Otsuka ELSZ-2000, Tokyo, Japan) and AsFlFFF (Wyatt Tech, Europe GmbH, Dernbach, Germany) were used.

The AsFlFFF channel was equipped with a cellulose membrane having the cut-off molecular weight (MW) of 10 kDa (Millipore, Bedford, MA, USA) and a 250 μm-thick Mylar spacer. The AsFlFFF carrier liquid was an aqueous solution containing 0.1% FL-70 and 0.02% sodium azide (NaN_3_). The carrier liquid flow was generated using a high-performance liquid chromatography (HPLC) pump (P-6000, FUTECS Co., Ltd., Daejeon, Republic of Korea). An Optiflow-1000 Liquid Flowmeter (Agilent Technologies, Palo Alto, CA, USA) was used to measure the flow rate. A UV detector (Spectra UV 150, Thermo Separation Products, Waltham, MA, USA) was used to monitor the CB particles being eluted from the AsFlFFF channel. All AsFlFFF analysis for the CB dispersions was performed at the channel flow rate and the cross-flow rate of 0.8 and 0.3 mL/min, respectively. The CB dispersion samples were injected using a syringe pump (Legato 110, KD Scientific Inc., Mendon, UT, USA) at the flow rate of 0.2 mL/min. The sample injection volume was 50 μL. To evaluate the reproducibility of the results, all the AsFlFFF measurements were repeated three times.

### 3.3. Synthesis of Styrene Maleic Acid (SMA) Copolymer Dispersants

SMA copolymer was synthesized as follows [35]. After adding 42 g of poly (styrene-co-maleic anhydride) (SMAnh) powder to a three-necked flask, 42 g of DI water was added and mixed by stirring on a heating mantle. When the temperature of the solution reached 80 °C, 24 g of ammonia water was added and mixed by stirring for 2 h to synthesize an SMA dispersant by a ring-opening reaction [36].

The use of three different types of SMAnh powder yielded three different types of SMA dispersants: ‘SMA-1000’, ‘SMA-2000’, and ‘SMA-3000’, respectively. The structures of the three different types of SMA’s prepared in this study are shown in Figure 4. The SMA-1000, SMA-2000, and SMA-3000 have 1, 2, and 3 styrene groups, respectively, in their repeating units.

SMA-1000, SMA-2000, and SMA-3000 are all in the form of off-shite flakes or powder. The molecular weight (Mw) is 5500, 7500, and 9500, respectively, and the more styrene structures there are, the higher the molecular weight. Polydispersity is expressed as the polydispersity index (PDI). The PDI is a criterion representing the breadth of molecular weight distribution and is defined as the ratio of weight average molecular weight (Mw) to the average molecular weight (Mn). The PDI of SMA-1000, SMA-2000, and SMA-3000 are 2.75, 2.50, and 2.50, respectively. SMA-1000 is the largest, and SMA-2000 and SMA-3000 have the same value.

### 3.4. Preparation of CB Dispersions

After mixing with 5 g of CB particles, 75 g of SMA dispersant, and 420 g of DI water, a CB dispersion was prepared using a basket mill (HSX0502251, Hyosung, Seoul, Republic of Korea). The size of the milling bead was 0.8 mm, and milling was performed at 1500 rpm using a total of 350 g of beads.

## 4. Results and Discussion

### 4.1. Determination of Storage Stability of CB Dispersions

The CB dispersions were prepared by basket-milling at the milling time of 1, 2, and 3 h using SMA-1000, SMA-2000, and SMA-3000, respectively, yielding nine CB dispersions in total. Then the CB dispersions were stored in an oven at 60 ℃ for 1, 7, and 11 days.

Figure 5, Figure 6 and Figure 7 show pH, viscosity, conductivity, and particle diameter measured for the CB dispersions prepared with SMA-1000, SMA-2000, and SMA-3000, respectively. Each of the CB dispersions was measured three times on days 1, 7, and 11 after the dispersions were prepared. The particle size was measured using DLS after 100-fold dilution of the dispersion. As mentioned earlier, all measurements were repeated three times.

In Figure 5, Figure 6 and Figure 7, the pH of the CB dispersions increases slightly for the first few days. Since ammonia is used in the preparation of the SMA dispersant, the pH of the dispersant is about 9.0. Usually, the surface of the carbon black particles is acidic (pH about 4.0). As the dispersant molecules are coated on the surface of carbon black particles, the pH increases, and the dispersion is stabilized as it approaches 9.0. The increase in pH found during the first few days after the preparation of the dispersions suggests it takes a few days for the CB dispersions to be stabilized. After about 7 days, pH tends to remain constant in all CB dispersions, indicating that, after about 7 days, the dispersants are well coated on the carbon black particles to stabilize the dispersions.

Viscosity also shows a similar tendency to pH, as shown in Figure 5b, Figure 6b, and Figure 7b. The viscosity tends to increase or decrease slightly during the first few days after dispersion preparation, remaining constant at ~1500 cp after about 7 days. In general, it is expected that high-viscosity CB dispersions will require a longer time to stabilize than low-viscosity dispersions.

In Figure 5, Figure 6 and Figure 7, the conductivity of all CB dispersions tends to decrease with increasing storage time. As the storage time increases, the SMA dispersant adsorbed on the surface of the CB particles may be desorbed, and the cohesive force between the particles gradually increases, thereby decreasing the particle surface charge, resulting in a reduction in the conductivity. For the CB dispersions made with SMA-1000, the conductivity reduction is greater (reduced by about half from about 800 to about 400 μs/cm) than the CB dispersions made with SMA-2000 or SMA-3000. Over time, it appears that the SMA-1000 dispersant desorbs more readily from the CB particle surface than the SMA-2000 or the SMA-3000, resulting in a greater reduction in the particle’s surface charge and, thus, a greater reduction in conductivity. This may be because the SMA-1000 dispersant is more easily desorbed from the CB surface during heat treatment due to the relative lack of styrene groups capable of covalently bonding to the CB surface. It seems difficult to obtain a stable dispersion of CB particles with the SMA-1000 dispersant.

All the CB dispersions showed average particle diameters of around 200 nm. As the styrene content of the dispersant increases from SMA-1000 to 3000, the change in the particle size with storage time decreases. It seems that as the styrene content of the dispersant increases, the adsorption of the dispersant on the carbon black particle surface becomes more stable. As shown in Figure 6 and Figure 7, in the case of CB-dispersion prepared with SMA-2000 or 3000, the particle sizes of the dispersions obtained by 2 h of milling were more stable than those of the dispersions obtained by 1 h or 3 h of milling. These results suggest that 1 h is insufficient to disperse the CB particles well, and 3 h of milling may cause over-dispersion, increasing the cohesive force (van der Waals force) between the particles.

The pH, viscosity, conductivity, and mean diameter measured after storing 11 days for the CB dispersions prepared with SMA-1000, 2000, and 3000 are summarized in Table 1, Table 2 and Table 3, respectively. All measurements were repeated three times, and the data shown in Table 1, Table 2 and Table 3 are expressed by ‘mean ± 1 standard deviation (SD)’.

As shown in Table 1, Table 2 and Table 3, all the CB dispersions have a pH of 8~9 and are weakly basic. As the milling time increases, the conductivity gradually decreases. With SMA-1000 or 2000, the viscosity and mean diameter of the CB dispersions decreased initially as the milling time increased and then increased after milling for 3 h. As mentioned earlier, it seems that milling for 3 h may cause overdispersion, and the cohesive force between particles increases, resulting in re-agglomeration of the CB particles. When the SMA-3000 dispersant was used, the particle size did not increase even after milling for 3 h, indicating that the dispersion was stable.

Figure 8 shows the change in viscosity according to the shear rate measured for three CB dispersions prepared with SMA-1000, SMA-2000, and SMA-3000. In all three dispersions, the viscosity decreases gradually as the shear rate increases. It is noted that the dispersions made with SMA-1000 and SMA-2000 showed higher viscosities than that made with SMA-3000 until the shear rate reached about 10 $s^{-1}$. When the particle size is reduced by milling, the specific surface area increases, which increases the van der Waals attraction between the particles, and the tendency of the particles to agglomerate with each other increases, and consequently, the viscosity increases. SMA-3000, which has more styrene groups, seems to bind more strongly to the surface of the CB particles than the SMA-1000 or 2000, better preventing the CB particles from rubbing against each other.

### 4.2. Particle Size Analysis of CB Dispersions

DLS and AsFlFFF were used to determine the particle size distribution of CB dispersions prepared with three different types of SMA dispersants. The measured samples were the CB dispersions stored at 60 °C for 11 days after preparation. Figure 9 shows the particle size distributions obtained by DLS for the CB dispersions prepared using SMA-1000 (a), SMA-2000 (b), and SMA-3000 (c), respectively. The mean particle diameters determined by DLS for the CB dispersions are shown in Figure 9 and Table 4. As mentioned earlier, all measurements were repeated three times, and the data shown in Table 4 are expressed by ‘mean ± 1 SD’. Table 4 also shows the relative standard deviation (RSD) determined by $\left(\frac{SD}{mean}\times 100\right)$.

In Figure 9a with SMA-1000, the particle size distribution narrows as the milling time increases. The mean particle size decreases from 472 to 340 after 2 h of milling. A slight increase in the mean particle size was observed after 3 h of milling from 340 to 359 nm, as shown in Table 4. In Figure 9b with SMA-2000, after 1 h of milling, the CB dispersion shows a bimodal size distribution with mean diameters of 257 and 1070 nm, respectively (see Table 4). As the milling time increases from 1 to 2 h, the particle size distribution narrows, and the average particle size decreases down to 252 nm, as shown in Figure 9a and Table 4. However, this time, after 3 h of milling, the particle size distribution broadens, and the particle size increases from 252 to 307 nm as shown in Table 4. In Figure 9c with SMA-3000, as the milling time increases, the particle size distribution narrows. The mean particle size decreases gradually from 390 down to 240 nm, as shown in Table 4.

All three CB dispersions showed trends of decreasing mean particle size and narrowing particle size distribution with increasing milling time after 2 h of milling. When the milling time was further increased up to 3 h, the CB dispersions did not show the same trend. The mean particle size of the CB dispersions either increases or decreases depending on the type of dispersant used to prepare the dispersions. The DLS results are shown in Figure 9 and Table 4, suggesting that the agglomeration of CB particles may occur due to over-dispersing after 3 h of milling. Based on observations so far, 2 h seems to be the optimal milling time.

The DLS data in Table 4 show relatively high SD values, suggesting that the repeatability of the DLS data is poor. The relatively poor repeatability of DLS makes the application of DLS difficult for the optimization of dispersing of CB particles.

Figure 10, Figure 11 and Figure 12 show AsFlFFF fractograms obtained for the CB dispersions prepared using SMA-1000, SMA-2000, and SMA-3000 (c), respectively. Additionally, shown at the right in Figure 10, Figure 11 and Figure 12 are the particle size distributions obtained from the fractograms using Equation (1).

Again, all measurements were repeated three times, and the mean particle sizes measured from the fractograms shown in Figure 10, Figure 11 and Figure 12 are summarized in Table 5 in the same manner as in Table 4 (mean diameter ±1 SD).

As shown in Figure 10, Figure 11 and Figure 12 and Table 5, when the SMA-1000 was used, the particle size distribution tended to narrow without significant changes in the mean particle size as the milling time increased. In the case of using the SMA-2000, the size of CB particles tended to decrease gradually as the milling time increased. When the SMA-3000 was used, the particle size and particle size distribution after 2 h of milling did not change much compared to those after 1 h of milling. After milling for 3 h, the particle size was increased. This indicates again that, with SMA-3000, 3 h of milling resulted in an increase in the particle size owing to over-dispersing.

It is noted that the relative standard deviation (RSD) values in Table 5 are lower than those in Table 4, which suggests that AsFlFFF provides higher precision in the measurement of CB particle size than DLS. In particular, for the CB dispersion prepared with the SMA-3000, AsFlFFF results showed RSD values of 8.4 and 9.2 after 1 and 2 h of milling, respectively, which were much lower than the DLS results (28.5 and 41.4).

The information on the CB particle size and the size distribution obtained from DLS and AsFlFFF suggests that the use of SMA-2000 or SMA-3000 yields more stable CB dispersions than the use of SMA-1000. When using SMA-2000 or SMA-3000, the milling time of 2 h seems to be optimal. When the SMA-2000 is used, the change in the CB particle size with the change in the milling time is relatively larger than when using the SMA-3000. With the SMA-3000, the CB particles have a relatively uniform particle size in the range of about 275 to 295 nm as the milling time was changed.

### 4.3. Colorimetric Analysis of CB Dispersions

Usually, the color or glossiness of particle dispersions changes with the particle size. The correlation between the color and the particle size in the CB dispersion was investigated. A colorimetric measurement provides the brightness ($L^{*}$), redness ($a^{*}$), and yellowness ($b^{*}$). Table 6 and Table 7 show the results from the colorimetric measurements of the CB dispersions prepared with SMA-2000 and SMA-3000, respectively, after being stored for 11 days at 60 ℃. The CB dispersion prepared with SMA-1000 was not measured as it was unstable. Again, all measurements were made three times

The colorimetric measurements showed that when the SMA-2000 was used, the lowest $L^{*}$ value was observed at the milling time of 3 h, whereas when the SMA-3000 was used, the lowest $L^{*}$ value was observed at the milling time of 1 h. The $L^{*}$ represents the brightness and luminosity. Lower $L^{*}$ value means the color is closer to black, and the particle size is smaller with a more homogeneous (or narrow) size distribution [37]. The trends in the $L^{*}$ values shown in Table 6 and Table 7 are consistent with the particle size results obtained from the AsFlFFF shown in Table 5. With the SMA-2000, the *L** value decreased as the milling time increased and reached the lowest (darkest) value of 23.9 after 3 h of milling. The AsFlFFF results also showed the smallest particle size (267 nm) after 3 h of milling, as shown in Table 5. With the SMA-3000, the $L^{*}$ value was the lowest after 1 h of milling, for which the AsFlFFF result showed the smallest and most narrow size distribution.

The $a^{*}$ and $b^{*}$ are measures of redness and yellowness, respectively. As shown in Table 6, with the SMA-2000, as the milling time increased, the redness did not change significantly, while the yellowness showed a tendency to increase. As shown in Table 7, with the SMA-3000, clear trends of redness or yellowness according to the milling time were not found.

## 5. Conclusions

The CB particles tend to agglomerate at elevated temperatures partly due to their relatively low thermal stability. Agglomeration of CB particles can cause color change and product defects. Selecting a stable dispersant and optimal dispersing time is thus highly important.

In the present study, the type of dispersant and milling time was selected to prepare stable CB dispersion through AsFlFFF and colorimetric analyses. In the particle size analysis of the CB dispersions, DLS results were less reproducible than AsFlFFF.

The results suggest that for the CB dispersion made in this study, the SMA-1000 is less useful than the SMA-2000 or the SMA-3000 because the use of SMA-1000 as a dispersant does not prevent the re-agglomeration of CB particles well. This is because the bonding of the SMA-1000 on the surface of the CB particles is relatively weak.

AsFlFFF and colorimetric analysis confirmed that stable CB dispersions could be obtained in 3 h of milling when using SMA-2000 or 1 h of milling when using SMA-3000.

The results of colorimetric and AsFlFFF analysis agree well, confirming that these analytical methods are powerful tools for selecting the dispersant type and milling time to disperse the CB particles and other types of particles.

## Figures and Tables

**Figure 1 polymers-14-05455-f001:**
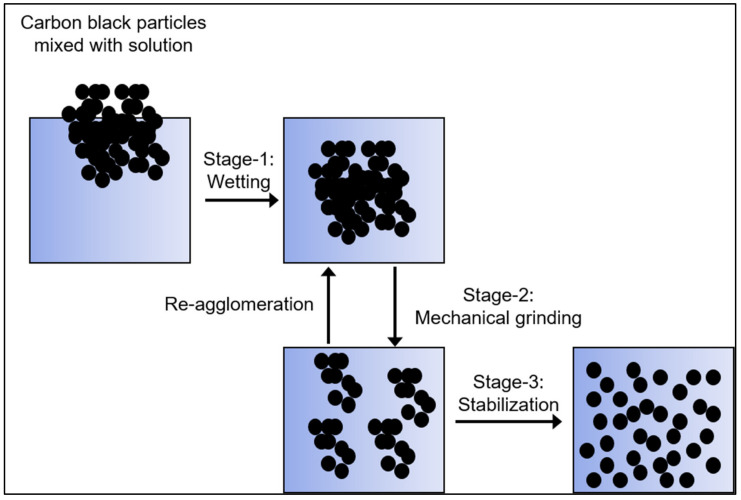
Three stages of CB particle-dispersing.

**Figure 2 polymers-14-05455-f002:**
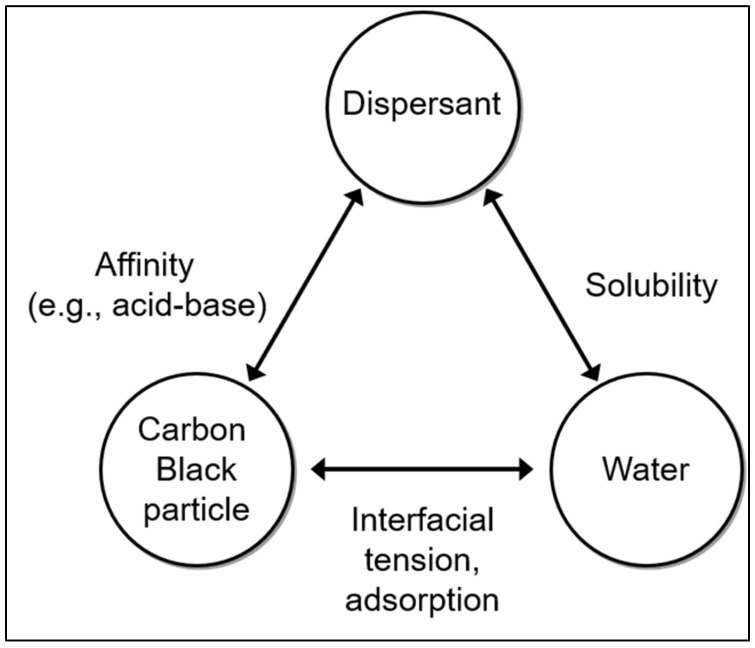
Three main components of aqueous CB dispersion and types of interactions affecting the dispersing of CB particles.

**Figure 3 polymers-14-05455-f003:**
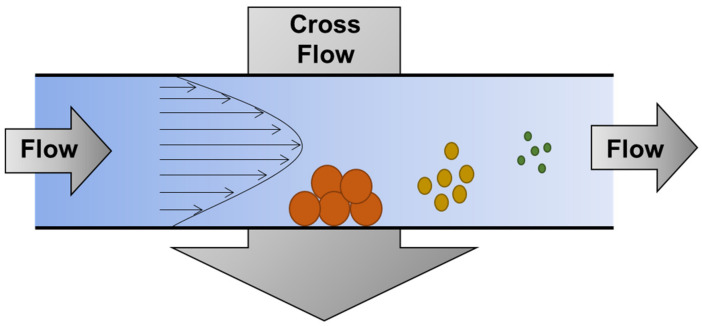
Scheme of asymmetrical flow field-flow fractionation (AsFlFFF).

**Figure 4 polymers-14-05455-f004:**
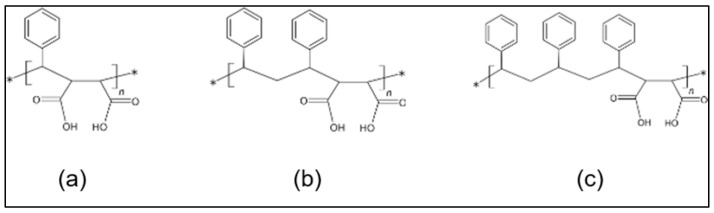
Structures of three types of SMA copolymer dispersants synthesized in this study; SMA-1000 (**a**), SMA-2000 (**b**), and SMA-3000 (**c**).

**Figure 5 polymers-14-05455-f005:**
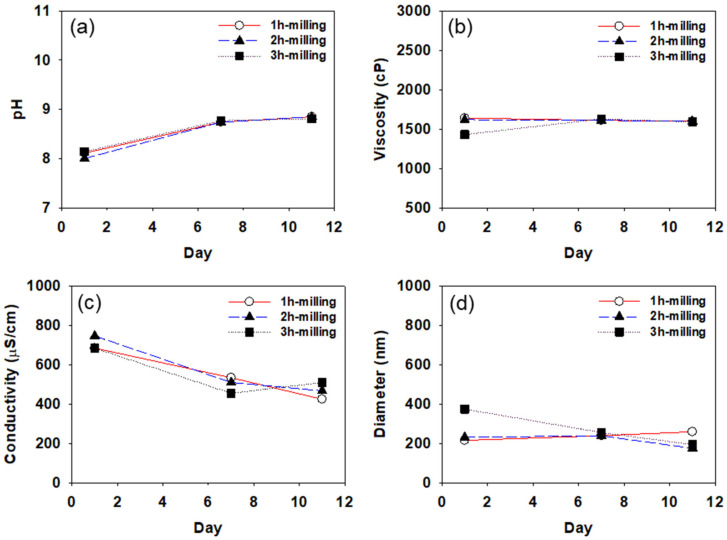
Variation of pH (**a**), viscosity (**b**), conductivity (**c**), and particle diameter (**d**) with time measured for the CB dispersions prepared using SMA-1000.

**Figure 6 polymers-14-05455-f006:**
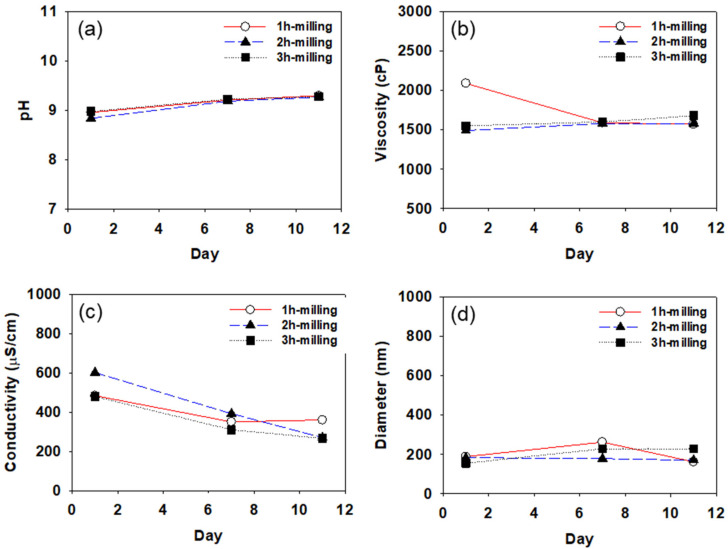
Variation of pH (**a**), viscosity (**b**), conductivity (**c**), and particle diameter (**d**) with time measured for the CB dispersions prepared using SMA-2000.

**Figure 7 polymers-14-05455-f007:**
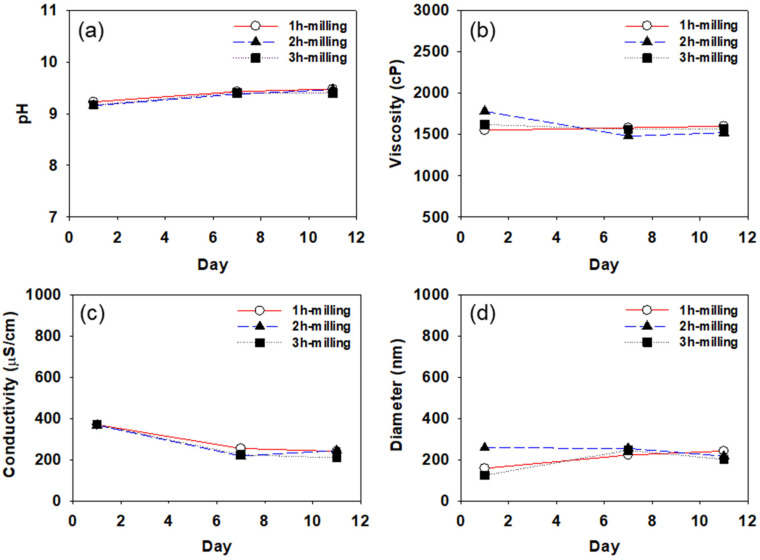
Variation of pH (**a**), viscosity (**b**), conductivity (**c**), and particle diameter (**d**) with time measured for the CB dispersions prepared using SMA-3000.

**Figure 8 polymers-14-05455-f008:**
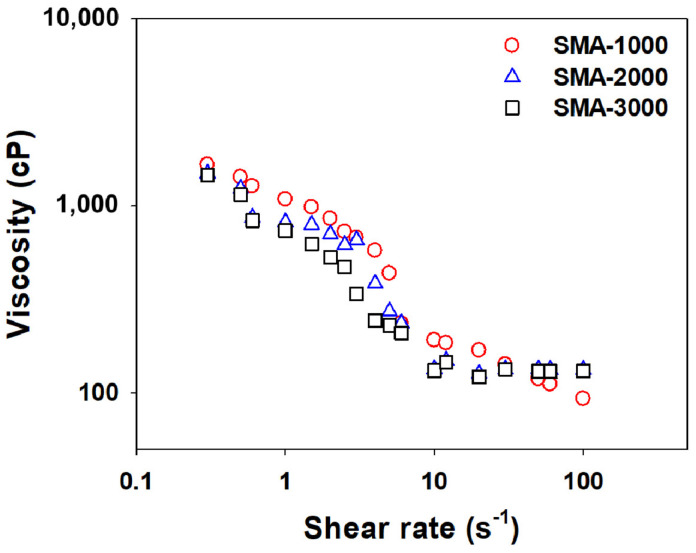
Variation of viscosity with shear rate measured for the CB dispersions prepared with SMA−1000 (**○**), SMA−2000 (**△**), and SMA−3000 (**□**), respectively.

**Figure 9 polymers-14-05455-f009:**
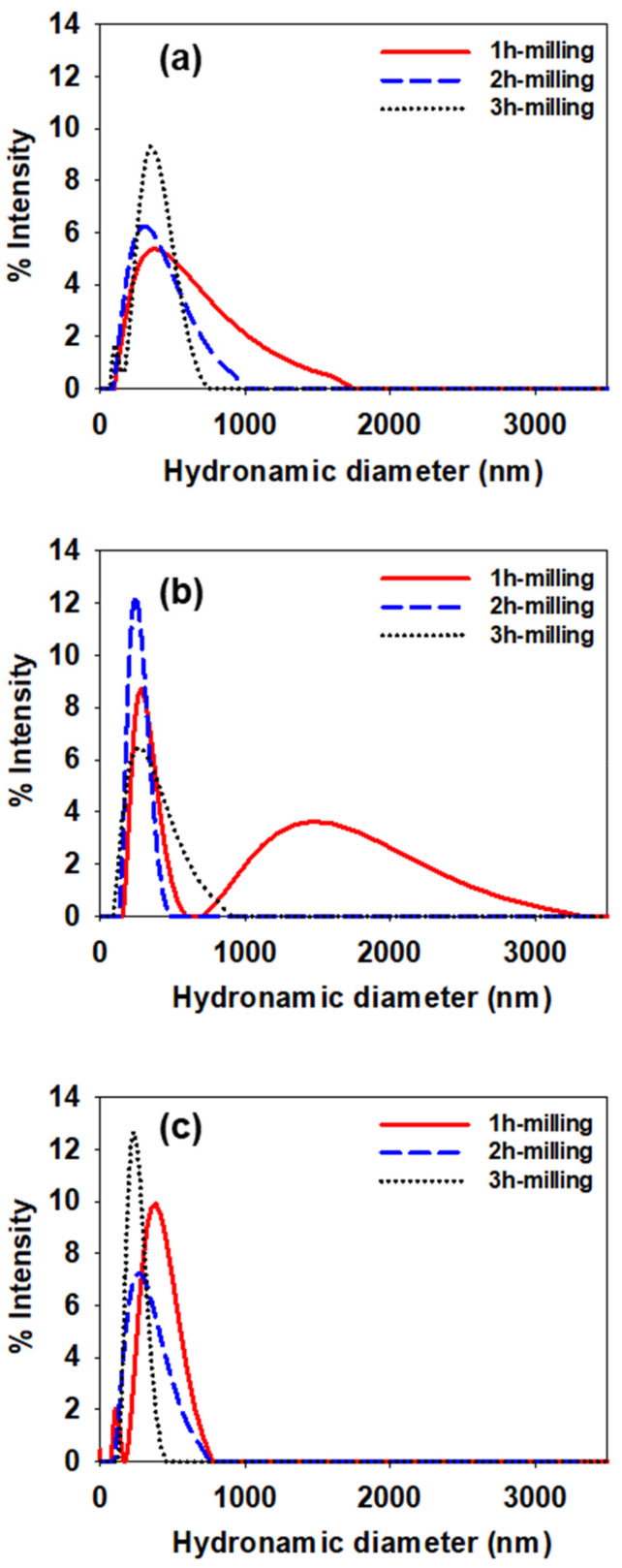
Particle size distributions obtained by DLS for the CB dispersions prepared using SMA-1000 (**a**), SMA-2000 (**b**), and SMA-3000 (**c**), respectively.

**Figure 10 polymers-14-05455-f010:**
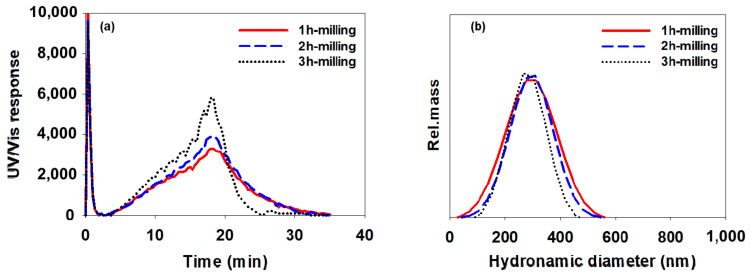
AsFlFFF fractograms (**a**) and size distributions (**b**) of the CB dispersions prepared with SMA-1000. The channel and the cross-flow rate were 0.8 and 0.3 mL/min, respectively. The carrier liquid was water containing 0.1% FL-70 and 0.02% NaN_3_.

**Figure 11 polymers-14-05455-f011:**
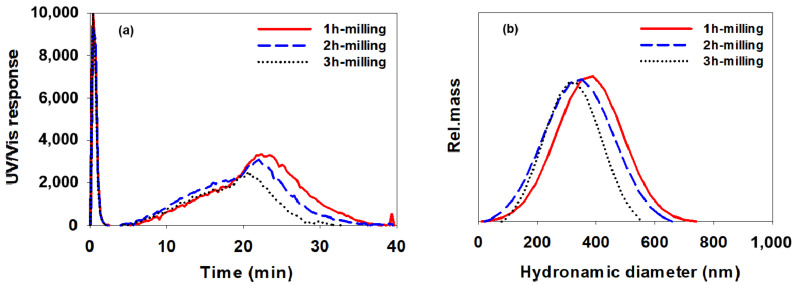
AsFlFFF fractograms (**a**) and size distributions (**b**) of the CB dispersions prepared with SMA-2000. All AsFlFFF conditions were the same as those in Figure 10.

**Figure 12 polymers-14-05455-f012:**
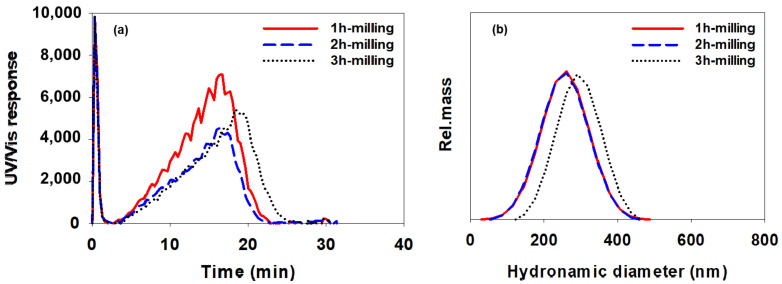
AsFlFFF fractograms (**a**) and size distributions (**b**) of CB dispersions prepared with SMA-3000. All AsFlFFF conditions were the same as those in Figure 10.

**Table 1 polymers-14-05455-t001:** pH, viscosity, conductivity, and diameter measured for CB dispersions prepared with SMA-1000 after storing 11 days.

Milling Time (h)	pH	Conductivity (μs/cm)	Viscosity (cP)	Diameter (nm)
1	8.46 ± 0.39	573 ± 118	1520 ± 199	238 ± 21.7
2	8.43 ± 0.42	593 ± 128	1510 ± 200	215 ± 34.7
3	8.45 ± 0.39	577 ± 112	1660 ± 232	274 ± 91.6

**Table 2 polymers-14-05455-t002:** pH, viscosity, conductivity, and diameter measured for CB dispersions prepared with SMA-2000 after storing 11 days.

Milling Time (h)	pH	Conductivity (μs/cm)	Viscosity (cP)	Diameter (nm)
1	9.14 ± 0.15	413 ± 168	1620 ± 358	204 ± 51.9
2	9.08 ± 0.19	462 ± 159	1450 ± 214	178 ± 65.6
3	9.12 ± 0.15	385 ± 111	1490 ± 241	204 ± 42.4

**Table 3 polymers-14-05455-t003:** pH, viscosity, conductivity, and diameter measured for CB dispersions prepared with SMA-3000 after storing 11 days.

Milling Time (h)	pH	Conductivity (μs/cm)	Viscosity (cP)	Diameter (nm)
1	9.14 ± 0.15	304 ± 66.6	1450 ± 259	207 ± 43.8
2	9.08 ± 0.19	294 ± 74.8	1470 ± 280	243 ± 22.6
3	9.12 ± 0.15	289 ± 84.5	1460 ± 258	190 ± 61.5

**Table 4 polymers-14-05455-t004:** Particle diameters determined by DLS for CB dispersions are shown in Figure 9.

Milling Time (h)	Particle Diameter (nm) of CB Dispersions Prepared with					
	SMA-1000		SMA-2000		SMA-3000	
	Mean ± SD	RSD	Mean ± SD	RSD	Mean ± SD	RSD
1	472 ± 292	61.7	257 ± 661,070 ± 335	25.731.3	390 ± 111	28.5
2	340 ± 168	49.4	252 ± 62	24.4	292 ± 121	41.4
3	359 ± 109	30.4	307 ± 146	47.5	240 ± 58.2	24.3

**Table 5 polymers-14-05455-t005:** Particle diameters determined by AsFlFFF for the CB dispersions prepared with three different types of SMA dispersants.

Milling Time (h)	Particle Diameter (nm) of CB Dispersions Prepared with					
	SMA-1000		SMA-2000		SMA-3000	
	Mean ± SD	RSD	Mean ± SD	RSD	Mean ± SD	RSD
1	291 ± 79.1	27.2	314 ± 90.0	28.7	274 ± 22.9	8.4
2	295 ± 77.1	26.1	279 ± 87.2	31.2	275 ± 25.3	9.2
3	286 ± 85.4	29.9	267 ± 46.0	17.2	295 ± 70.3	23.8

**Table 6 polymers-14-05455-t006:** Colorimetric results of the CB dispersion prepared with SMA-2000.

Milling Time (h)	$L^{*}$	$a^{*}$	$b^{*}$
1	24.3 ± 0.01	0.34 ± 0.03	0.08 ± 0.02
2	24.3 ± 0.01	0.34 ± 0.02	0.33 ± 0.04
3	23.9 ± 0.01	0.33 ± 0.04	0.28 ± 0.02

**Table 7 polymers-14-05455-t007:** Colorimetric results of the CB dispersion prepared with SMA-3000.

Milling Time (h)	$L^{*}$	$a^{*}$	$b^{*}$
1	23.3 ± 0.01	0.38 ± 0.06	0.23 ± 0.02
2	24.6 ± 0.01	0.29 ± 0.04	0.12 ± 0.02
3	24.1 ± 0.01	0.31 ± 0.03	0.25 ± 0.02

## Data Availability

Not applicable.

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
