# Peer review of "A Study on Aqueous Dispersing of Carbon Black Nanoparticles Surface-Coated with Styrene Maleic Acid (SMA) Copolymer"

_polymers, 2022, doi:10.3390/polym14245455_

Round 1

Reviewer 1 Report

This is high proffesional work studying the stabilization of CB dispersions by means of polymeric dispersant SMA 1000 - SMA 3000 and the time of milling. The authors aimed at finding the optimal conditions to prevent the aglomeration of the CB particles. They used for this study DLS, AsFiFFF, viscosimetry, and optometry to determine which is the best dispersion and find out that SMA 2000 and SMA 3000 and milling time 1 h and 2 h have the best effect. So, they found the optimal conditions. I have some minor requirements prior publishing:

1. I will appreciate if they present as scheme of AsFIFFF method, as far as this is new method and the readers will enjoy to know something new;

2. There is repeating of text in Chapter 2 (AsFIFFF theory). The repeating should be removed.

3. And finally I would suggest them in their next work to use microbubbles as dispesant in presence of PPG 600. I think they will get good results.

Finally, I recommend publication after the minor revision.

Author Response

  1. I will appreciate if they present as scheme of AsFIFFF method, as far as this is new method and the readers will enjoy to know something new;

Add answer : Thank you for your comment. As part of the AsFIFFF theory, we have included a scheme of AsFlFFF as shown in Figure 3. (Page 4, AsFlFFF theory part modified in red (Please see the attachment))

  1. There is repeating of text in Chapter 2 (AsFIFFF theory). The repeating should be removed.

Add answer : Thanks for your detail comments. The repetitive parts have been deleted. (Page 4, AsFlFFF theory part modified in red (Please see the attachment))

  1. And finally I would suggest them in their next work to use microbubbles as dispesant in presence of PPG 600. I think they will get good results.

Finally, I recommend publication after the minor revision.

Add answer : Thanks for the great suggestion. We will try to obtain good results by referring to the suggestions for the next study.

Reviewer 2 Report

This paper aims at characterizing the aqueous dispersing of carbon black (CB) nanoparticles surface-coated with styrene maleic acid (SMA) copolymer. The asymmetrical flow field-flow fractionation (AsFlFFF) and dynamic light scattering (DLS) were used to measure the particle size distributions of the aqueous CB dispersions. However, there are two major issues which prevent publication in Polymers.

The aim of this study is not clear. This paper doesn't carry significant novelty.

The second problem deals with data analysis of the storage stability of CB dispersions. The authors have measured the variation of pH, viscosity, conductivity and particle diameter with time for CB dispersions prepared with different SMA dispersants. However, the results and explanation are not convincing. For example, as show in Tables 1-3, why the particle size decreases first and then increases with increasing milling time for CB dispersions prepared with SMA-1000 and SMA-2000? Why the trend is different for the aqueous CB dispersions prepared with SMA-3000? The discussion and understanding are insufficient. The reason needs to be clarified.

Minor points:

(1) They should provide a precise characterization of SMA dispersants including molecular weight and polydispersity.

(2) An additional control without SMA dispersant should be included in the comparisons of CB dispersions.

Author Response

This paper aims at characterizing the aqueous dispersing of carbon black (CB) nanoparticles surface-coated with styrene maleic acid (SMA) copolymer. The asymmetrical flow field-flow fractionation (AsFlFFF) and dynamic light scattering (DLS) were used to measure the particle size distributions of the aqueous CB dispersions. However, there are two major issues which prevent publication in Polymers.

The aim of this study is not clear. This paper doesn't carry significant novelty.

Add answer : Thank you for your comment. The optimization of particle-dispersing process includes the selection of proper dispersants the amount of dispersant, milling speed, milling time, and the development of dispersing method. Among them, selecting an appropriate dispersant for an aqueous dispersion of CB particles or finding optimal dispersion conditions (e.g., mechanical dispersion time) is not straightforward. If the mechanical dispersion time (e.g., milling time) is not sufficient, the particle size distribution tends to be broad due to the presence of agglomerated particles. Conversely, if the milling time is too long ('over-dispersing'), the specific surface area of the CB particles increases as the particle size decreases, which increases the van der Waals attraction between the CB particles, leading to agglomeration of the particles .

In this study, the stabilization of CB dispersions by means of polymeric dispersant SMA 1000 - SMA 3000 and the time of milling. In particular, aimed at finding the optimal conditions to prevent the aglomeration of the CB particles. The prepared SMA dispersant was strongly coated on the surface of the CB particles, and it was confirmed whether sufficient steric hindrance existed between the CB particles. Used for this study DLS, AsFiFFF, viscosimetry, and optometry to determine which is the best dispersion and find out that SMA 2000 and SMA 3000 and milling time 1 h and 2 h have the best effect. Also, the effect of the type of dispersant and milling time on the CB dispersion were investigated. AsFlFFF and colorimetric analysis confirmed that stable CB dispersions could be obtained in 3 h of milling when using SMA-2000 or 1 h of milling when using SMA-3000.

The second problem deals with data analysis of the storage stability of CB dispersions. The authors have measured the variation of pH, viscosity, conductivity and particle diameter with time for CB dispersions prepared with different SMA dispersants. However, the results and explanation are not convincing. For example, as show in Tables 1-3, why the particle size decreases first and then increases with increasing milling time for CB dispersions prepared with SMA-1000 and SMA-2000?

Add answer : Thank you for your comment. In general, as milling time increases, particle size decreases. However, if the milling time is too long ('over-dispersing'), the specific surface area of the CB particles increases as the particle size decreases, which increases the van der Waals attraction between the CB particles, leading to agglomeration of the particles . As a result, the particle size increases.

Why the trend is different for the aqueous CB dispersions prepared with SMA-3000? The discussion and understanding are insufficient. The reason needs to be clarified.

Add answer : Thank you for your comment. The prepared SMA-3000 dispersant is a copolymer with the largest number of hydrophobic styrene groups (Figure 4), and is strongly coated on the hydrophobic surface of CB particles, so that sufficient steric hindrance exists between CB particles. Therefore, the particle size decreased despite the increase in milling time.

Minor points:

(1) They should provide a precise characterization of SMA dispersants including molecular weight and polydispersity.

Add answer : Thank you for your comment. Added information on molecular weight and polydispersity for SMA dispersants. (Page 5. 3.3. Synthesis of styrene maleic acid (SMA) copolymer dispersants part added in red (see attachment)

(2) An additional control without SMA dispersant should be included in the comparisons of CB dispersions.

Add answer : Thanks for your detail comments.. CB particles cannot be dispersed in water without a dispersant because the surface itself is hydrophobic. Therefore, it is impossible to prepare a control solution without the SMA dispersant, so please understand that there is no control comparision data.

Reviewer 3 Report

The manuscript entitled “A study on aqueous dispersing of carbon black nanoparticles surface-coated with styrene maleic acid (SMA) copolymer” presented the application of styrene maleic acid (SMA) copolymer as a carbon black nanoparticles dispersant. The effect of the type of dispersant and milling time on the CB dispersion were investigated. AsFlFFF and colorimetric analysis confirmed that stable CB dispersions could be obtained in 3 h of milling when using SMA-2000 or 1 h of milling when using SMA-3000. In general, the concept of the present work is interesting. The overall experimental design was reasonable and the content was substantial. The conclusion was basically supported by the findings. In my opinion, the manuscript can be published after major revisions.
1. The molecular weight of dispersant have a significant effluence on CB dispersion. More information regarding the molecular weight of SMA should be provided in this work.

2. The amount of dispersant and milling speed should have a significant effluence on CB dispersion. If possible, more information regarding the effect of amount of dispersant and milling speed on CB dispersion, should be provided in the present work, which will make the paper more readable.

3. The dispersion mechanism of CB coated with SMA should be further discussed, which is beneficial for reader to understand this work. 

4. The figures of the paper need use clear expressions. Such as, in Figure 6 and Figure 8, different curves may be expressed in different colors.

5. Pay attention to the format of references. Such as, the writing format of journal article titles should be unified in References, capitalize the first word of a sentence or each word

(9) S. Lee.; C.H. Eum.; W.J. Kim. Surface Modification of Carbon Black Using Polymer Resin Synthesized by a Phenyl Radical Reaction. J. Korean Chem. Soc. 2016, 60, 286–291.

(10) W. Kim.; J. Bae.; C.H. Eum.; J. Jung.; S. Lee. Study on dispersibility of thermally stable carbon black particles in ink using asymmetric flow field-flow fractionation (AsFlFFF). Microchem. J. 2018, 142, 167–174.

Author Response

  1. The molecular weight of dispersant have a significant effluence on CB dispersion. More information regarding the molecular weight of SMA should be provided in this work.

Add answer : Thank you for your comment. Added information on molecular weight and polydispersity for SMA dispersants. (Page 5, 3.3. Synthesis of styrene maleic acid (SMA) copolymer dispersants part added in red (Please see the attachment))

  1. The amount of dispersant and milling speed should have a significant effluence on CB dispersion. If possible, more information regarding the effect of amount of dispersant and milling speed on CB dispersion, should be provided in the present work, which will make the paper more readable.

Add answer : Thank you for your detail comments. For readability, effect of dispersant amount and milling speed were added to the introduction when optimizing the CB dispersion process.(Page 3, Introduction part added in red (Please see the attachment))

  1. The dispersion mechanism of CB coated with SMA should be further discussed, which is beneficial for reader to understand this work.

Add answer : Thank you for your comment. Added dispersion mechanism of CB coated with SMA.(Page 3, Introduction part added in red (Please see the attachment))

  1. The figures of the paper need use clear expressions. Such as, in Figure 6 and Figure 8, different curves may be expressed in different colors.

Add answer :  Thank you for your comment. Based on the comments, I added color along the lines to all pictures, not just Figure 6 and Figure 8, to increase visibility.(all Figures added in red (Please see the attachment))

  1. Pay attention to the format of references. Such as, the writing format of journal article titles should be unified in References, capitalize the first word of a sentence or each word

(9) S. Lee.; C.H. Eum.; W.J. Kim. Surface Modification of Carbon Black Using Polymer Resin Synthesized by a Phenyl Radical Reaction.J. Korean Chem. Soc. 2016, 60, 286–291.

(10) W. Kim.; J. Bae.; C.H. Eum.; J. Jung.; S. Lee. Study on dispersibility of thermally stable carbon black particles in ink using asymmetric flow field-flow fractionation (AsFlFFF).Microchem. J.2018, 142, 167–174.

Add answer :  Thank you for your comment. We unified the title format of the papers by capitalizing the first word or each word in a sentence. (Page 15-16. reference part added in red(Please see the attachment))

Round 2

Reviewer 2 Report

This manuscript can be accepted.

Reviewer 3 Report

In general, the revision work is extensive and convincing. Thus, I would like to recommend the manuscript for publication.